# Enantiodivergent Synthesis of Benzoquinolizidinones from L-Glutamic Acid

**DOI:** 10.3390/molecules26195866

**Published:** 2021-09-28

**Authors:** Punlop Kuntiyong, Duangkamon Namborisut, Kunita Phakdeeyothin, Rungrawin Chatpreecha, Kittisak Thammapichai

**Affiliations:** Department of Chemistry, Faculty of Science, Silpakorn University, Muang Nakhon Pathom 73000, Thailand; duangkamon177@gmail.com (D.N.); kunita.phakdeeyothin@gmail.com (K.P.); chatpreecha_R@su.ac.th (R.C.); kittisakthammapichai@gmail.com (K.T.)

**Keywords:** enantiodivergent, benzoquinolizidinone, L-glutamic acid

## Abstract

Benzoquinolizidinone systems were synthesized in both enantiomeric forms from L-glutamic acid. The key chiral arylethylglutarimide intermediate was synthesized from dibenzylamino-glutamate and homoveratrylamine. Aldol reaction of the glutarimide afforded a mixture of *syn* and *anti*-aldol adducts. Subsequent regioselective hydride reduction of the glutarimide carbonyl followed by *N*-acyliminium ion cyclization afforded a product with opposite absolute configurations at C3 and C11b. Cope elimination of the dibenzylamino group then converted the two diastereomers into enantiomers.

## 1. Introduction

As a subfamily of tetrahydoisoquinoline, benzoquinolizidine has a fused piperidine as the third ring. It is an important structural motif found in various alkaloids, either natural or synthetic molecules. Many of these alkaloids possess interesting biological activities [1,2,3]. For examples, protoemetinol (**1**) is a natural tetrahydroisoquinoline alkaloid which is found as a substructure or biosynthetic precursor of other biologically active alkaloids such as emetine (**2**), an anti-amoebic substance which also inhibits breast tumor growth [4,5]. Tetrabenazine (**3**) is a synthetic drug for treatment of motor function disorder due to Huntington’s disease [6]. Notably, protoemetinol and tetrabenazine have opposite configuration at C11b. Other benzoquinolizidine alkaloids include alpha-glucosidase inhibitors schulzeines and alangine (**4**) (Figure 1) [7,8].

Several procedures for the synthesis of benzoquinolizidine frameworks have been reported both as racemic and enantioselective synthesis [9]. Bischler–Napieralski and Pictet–Spengler are common reactions in many tetrahydroisoquinoline syntheses [10,11,12]. There have been impressive routes featuring tandem or cascade condensation/cyclization that construct the tricyclic benzoquinolizidinone core in a single step [13]. A few reported asymmetric syntheses employ tyrosine-based chiral starting material for diastereoselective synthesis of the tetrahydroisoquinoline core (Figure 1) [14]. An enantioselective Pictet–Spengler reaction of tryptamine using chiral thiourea and binaphthol-derived phosphoric acid has been reported. However, an enantioselective Pictet–Spengler reaction of arylethylamine is less common. Chiral sulfoxide and Oppolzer’s sultam were used as a chiral auxiliary for such reaction, as reported by Koomen [15] and Czarnocki [16], respectively, while enantioselective syntheses featuring asymmetric hydrogenation and allylation of dihydroisoquinoline intermediates have also been reported [17].

We have previously reported a diastereoselective synthesis of benzoquinolizidinone and indoloquinolizidinone cores using cyclization of chiral *N*-acyliminium ion derived from L-glutamic acid and arylethylamine and tryptamine, respectively [18]. The amine was coupled with benzyl *N,N*-dibenzylglutamate to give the corresponding amide. Subsequent treatment with lithium aluminum hydride gave the arylethylglutarimide **7** and the corresponding hydroxylactam, which was the precursor of the *N*-acyliminium ion. Conversion of glutarimide **7** to the hydroxylactam was then completed using DIBALH reduction. Subsequently, we synthesized glutarimide **7** with a modified route, where homoveratrylamine was reacted with dimethyl *N,N*-dibenzylglutamate **6** in the presence of LDA in THF to give glutarimide **7** in good yield. DIBALH reduction of the less hindered carbonyl gave the corresponding hydroxylactam, which upon treatment with TMSOTf gave benzoquinolizidinone **8** in a 5.7:1 diastereomeric ratio. The relative configurations were determined by NOESY experiments which showed correlation between H11b and H3. Cope elimination of the dibenzylamino group gave tricyclic enamide **9** (Figure 2). This functionality is suitable for further manipulation to install additional substituents via Michael and hetero-Michael addition. 

The C11b configuration (*S*) derived from axial attack of the six-membered *N*-acyliminium ion with the dibenzylamino group adopting a pseudo-equatorial position imposing the stereo-control. Although the functionalized benzoquinolizidinones were obtained in good overall yield and respectable diastereoselectivity, achieving opposite configuration at the C11b using the same procedure would require starting the route with less accessible D-glutamic acid. In order to be able to arrive at both enantiopodes in a non-racemic form from L-glutamic acid, we devised an alternative approach.

## 2. Results and Discussion

To alter the mode of stereochemical control, we envisioned that installing an additional substituent on the glutarimide ring would force new modes of stereo-control. This was achieved by aldol reaction at α-methylene upon treatment of glutarimide **7** with LDA and acetaldehyde. We envisioned that the additional substituent would also control the regioselectivity of the subsequent hydride reduction of the glutarimide carbonyl. To maximize this effect, the aldol adduct would be converted to the corresponding TBS-silyl ether. The aldol product was obtained as a mixture of diastereomers which was separable by chromatography only after treatment of the aldol adduct **10** with TBSOTf to give TBS ethers **11** and **12** in a 2:1 diastereomeric ratio. The kinetic aldol adduct **11** was obtained as the major product (Figure 3). 

As expected, DIBALH reduction of glutarimide **11** was exclusive at the carbonyl adjacent to the stereocenter bearing the dibenzylamino group, which is different from the reduction of glutarimide **7**. Without purification, the resulting hydroxylactam **13** was treated with TMSOTf to give tricyclic benzoquinolizidinone **14** as a single diastereomer with concomitant removal of the TBS group. The relative configuration of the tricyclic product was determined by NOESY experiments showing correlation between H11b and H3. Cope elimination then gave the functionalized benzoquinolizidinone **15** with a substituent on the C3 position (Figure 4).

The aldol adduct **12** underwent DIBALH reduction and subsequent *N*-acyliminium ion cyclization to give tricyclic benzoquinolizidinone **17** via hydroxylactam **16**. Cope elimination of the dibenzylamino group gave the product that has identical ^1^H and ^13^C NMR spectra to those of benzoquinolizidinone **15** (Appendix A). However, the optical rotations of the two compounds are similar in magnitude, with opposite signs (+32.8 and −44.8, respectively) (Figure 5). Therefore, they are enantiomers. The discrepancy between the optical rotations of benzoquinolizidinones **15** and *ent*-**15** may be resulted from impurity, observable in the aliphatic region in the 1H NMR spectrum, in the sample of benzoquinolizidinone **15** used for the measurement. The results were replicable in the reaction sequence beginning with the aldol reaction of glutarimide **7** and isobutyraldehyde.

The aldol reaction of glutarimide **7** with isobutyraldehyde in the presence of LDA in THF gave a mixture of aldol adducts **18** and **19**, which were separable by chromatography in a ratio of 1.9:1 (Figure 6). We discovered that conversion to silyl ether was not necessary for either separation of the diastereomeric products or for control of regioselective reduction of the glutarimide. 

The DIBALH reduction of glutarimide **18** occurred at the carbonyl adjacent to the stereocenter bearing the dibenzylamino group to give hydroxylactam **20**, which was treated with TMSOTf directly without purification. The product tricyclic benzoquinolizidinone **21** was obtained as a single diastereomer. Cope elimination then gave the functionalized benzoquinolizidinone **22** (Figure 7).

The aldol adduct **19** underwent DIBALH reduction and subsequent *N*-acyliminium ion cyclization to give tricyclic benzoquinolizidinone **24** via hydroxylactam **23**. Cope elimination of the dibenzylamino group gave the product that has identical ^1^H and ^13^C NMR spectra to **22**. However, the optical rotations of the two compounds are similar in magnitude but have opposite signs (+83.7 and −79.5) (Figure 8). Therefore, they are enantiomers **22** and *ent*-**22**.

The relative configuration of the benzoquinolizidinones was determined by NOESY experiments. Strong through-space correlation was observed between the H11b and H3 and between H1 and H3. The rationale for the stereochemical outcome of the *N*-acyliminium ion cyclization can be appreciated by considering the transition state of the reaction. The *N*-acyliminium ion from hydroxylactams **16** and **23** with *syn*-relative configurations of the substituents on the glutarimide ring adopted the half-chair conformation in which the substituents assumed the pseudo-equatorial positions (***TS1***). The nucleophilic dimethoxybenzene ring attacked from the axial position of this *N*-acyliminium ion, giving the product with *S*-configuration at C11b (Figure 9). 

On the other hand, the *N*-acyliminium ion from hydroxylactams **13** and **20** with *anti*-relative configurations of the substituents on the glutarimide ring adopted the twisted-boat conformation, which created a convex/concave bias between the two diastereotopic faces of the transition state (***TS2***). The nucleophilic dimethoxybenzene ring attacked from the convex side of the *N*-acyliminium ion, giving the product with *R*-configuration at C11b (Figure 10). Strong through-space correlation was observed between the H11b and H3 in the NOESY spectra.

The *R/S* ratio in this route was in favor of *R* for the *N*-acyliminium ion cyclization product in moderate stereoselectivity. It is possible to obtain products with all *S* configurations at the C11b if the aldol reaction gives the products with 2,5-*syn*-disubstituted glutarimide. In this manner, glutarimide **7** was treated with three equivalents of NaHMDS followed by acetaldehyde. This resulted in isomerization of the initially formed kinetic product (as in **11**) to give the aldol adducts **26** and **12** after silylation of the secondary alcohol. Subsequent reduction with DIBALH of glutarimide **26** and treatment with TMSOTf resulted in the *N*-acyliminium ion cyclization product **28** with the *S* configuration at C11b via hydroxylactam **27** (Figure 11).

## 3. Experimental Section

Starting materials and reagents were obtained from commercial sources and were used without further purification. Solvents were dried by distillation from the appropriate drying reagents immediately prior to use. Tetrahydrofuran and ether were distilled from sodium and benzophenone under argon. Toluene, triethylamine and dichloromethane were distilled from calcium hydride under argon. Moisture- and air-sensitive reactions were carried out under an atmosphere of argon. Reaction flasks and glassware were oven-dried at 105 °C overnight. Unless otherwise stated, concentration was performed under reduced pressure. Analytical thin-layer chromatography (TLC) was conducted using Fluka precoated TLC plates (0.2 mm layer thickness of silica gel 60 F-254). Compounds were visualized by ultraviolet light and/or by heating the plate after dipping in a 1% solution of vanillin in 0.1M sulfuric acid in EtOH. Flash chromatography was carried out using Scientific Absorbents Inc. silica gel (40 mm particle size). Optical rotations were measured with a Perkin-Elmer 243 polarimeter at ambient temperature using a 1 dm cell with 1 mL capacity. Infrared (IR) spectra were recorded on a Nicolet 5DXB FT-IR spectrometer. Proton and carbon nuclear magnetic resonance (NMR) spectra were obtained using a Bruker Avance-300 spectrometer.

### 3.1. (3S, 5R)- and (3S,5S)-1-(3,4-Dimethoxyphenylethyl)-3-(dibenzylamino)-5-(1-hydroxyethyl) Piperidine-2,6-dione TBS Ethers **11** and **12**

To a solution of glutarimide **7** (787 mg, 1.7 mmol) in dry THF (10 mL), LDA (2M solution in THF, 2.50 mL, 2.5 mmol) was added at −78 °C under an argon atmosphere, and the solution was stirred for 15 min. Acetaldehyde (0.3 mL, 5 mmol) was added to the mixture and the solution was stirred for an additional hour. The reaction was quenched by adding sat. aq. NH_4_Cl (10 mL) into the reaction. The mixture was extracted with CH_2_Cl_2_ (3 × 10 mL). The combined organic layers were dried over anhydrous Na_2_SO_4_, filtered and evaporated under reduced pressure. Purification of the crude material by flash column chromatography (silica gel, 6:1 hexane/ethyl acetate) produced the aldol adducts as a mixture of diastereomers. This mixture was dissolved in dry CH_2_Cl_2_ (10 mL). To this solution, 2,6-lutidine (0.20 mL, 1.8 mmol) and TBSOTf (0.44 mL, 1.8 mmol) were added at 0 °C under an argon atmosphere, and the solution was stirred for 3 hours. The reaction was quenched with sat. aq. NaHCO_3_ (20 mL) and extracted with CH_2_Cl_2_ (3 × 20 mL). The combined organic layers were dried over anh. Na_2_SO_4_, filtered and evaporated under reduced pressure. The crude product was purified by column chromatography (silica gel, 4:1 hexane/EtOAc) to give silyl ethers (**11** = 481 mg, 46%, **12** = 241 mg, 23%) as yellow oils.

**11**: R_f_ (4:1 hexane/EtOAc) 0.60; ^1^H NMR (300 MHz, CDCl_3_) δ 7.45–7.19 (m, 10H), 6.81–6.64 (m, 3H), 4.18–4.10 (m, 1H), 4.00–3.70 (m, 4H), 3.89 (s, 3H), 3.80 (s, 3H), 3.60 (d, *J* = 13.1 Hz, 2H), 3.47 (ddd, *J* = 12.2, 7.0, 3.1 Hz, 1H), 2.90–2.82 (m, 1H) 2.72 (t, *J* = 7.0 Hz, 2H), 2.30–1.95 (m, 2H), 1.05 (d, *J* = 7.0 Hz, 3H), 0.79 (s, 9H), −0.01 (s, 3H), −0.20 (s, 3H); ^13^C NMR (75 MHz, CDCl_3_) δ 173.2, 172.3, 148.8, 147.6, 139.5(2C), 130.7, 128.4(4C), 128.3(4C), 127.1(2C), 112.3, 110.9, 66.7, 59.8, 55.75, 55.7(2C), 54.8, 48.9, 40.7, 33.6, 25.7(3C), 20.9, 18.6, 17.9, −4.6, −5.2; ${\left[\alpha \right]}_{25}^{D}$ +40.0 (c 1.0, CHCl_3_); νmax (film) 2950, 2929, 1724, 1669, 1515, 1260, 1028, 775 cm^−1^; ESI-HRMS calculated for C_37_H_50_N_2_NaO_5_Si [M + Na]^+^ 653.3387, found 653.3395.

**12**: R_f_ (4:1 hexane/EtOAc) 0.51; ^1^H NMR (300 MHz, CDCl_3_) δ 7.46–7.19 (m, 10H), 6.84–6.69 (m, 3H), 4.78–4.66 (m, 1H), 3.98 (d, *J* = 13.1 Hz, 2H), 4.00–3.80 (m, 2H), 3.89 (s, 3H), 3.79 (s, 3H), 3.75 (d, *J* = 13.2 Hz, 2H), 3.49–3.41 (m, 1H), 2.75 (dt, *J* = 7.1, 2.3 Hz, 2H), 2.30–2.10 (m, 2H), 2.10–1.96 (m, 1H), 1.20 (d, *J* = 7.0 Hz, 3H), 0.85 (s, 9H), 0.10 (s, 3H), 0.05 (s, 3H); ^13^C NMR (75 MHz, CDCl_3_) 173.3, 172.4, 148.8, 147.6, 139.8, 139.5, 131.2, 128.6(2C), 128.5(2C), 128.4(2C), 128.2(2C), 127.1(2C), 121.0, 112.2, 111.0, 66.6, 58.8, 55.9, 55.8, 54.9, 49.2, 41.6, 33.9, 25.8(3C), 25.7, 21.7, 21.3, 17.9, −4.3, −5.0;$\;{\left[\alpha \right]}_{25}^{D}$ −10.0 (c 1.0, CHCl_3_) ); νmax (film) 2954, 2928, 1725, 1675, 1516, 1261, 1028, 764 cm^−1^; ESI-HRMS calculated for C_37_H_51_N_2_O_5_Si [M + H]^+^ 631.3567, found 631.3548.

### 3.2. (1S,3R,11bR)-1-(Dibenzylamino)-2,3,6,7-tetrahydro-3-((S)-1-hydroxyethyl)-9,10-dimethoxy-1H-pyrido[2,1-a]isoquinolin-4(11bH)-one (**14**)

To a solution of imide **11** (85 mg, 0.14 mmol) in dry toluene (10 mL), DIBALH (0.28 mL, 1M in toluene, 0.28 mmol) was added under an argon atmosphere at −78 °C. Upon completion adjudged by TLC (1 h), the reaction was quenched with MeOH (2 mL). The resulting mixture was allowed to warm to room temperature and sat. aq. NaHCO_3_ (10 mL) was added. The mixture was extracted with EtOAc (3 × 10 mL) and the combined organic layer were dried over anh. Na_2_SO_4_, filtered and concentrated under reduced pressure to give the hydroxylactam **13** as a yellow oil. This compound was dissolved in dry CH_2_Cl_2_ (10 mL), and to the resulting solution, TMSOTf (0.1 mL, 0.42 mmol) was added at 0 °C under an argon atmosphere and the solution was stirred for 3 hours. The reaction was quenched with sat. aq. NaHCO_3_ (10 mL) and extracted with CH_2_Cl_2_ (3 × 10 mL). The combined organic layers were dried over anh. Na_2_SO_4_, filtered and evaporated under reduced pressure. The crude product was purified by flash chromatography (silica gel, 4:1 hexane/EtOAc) to give dibenzylamino-benzoquinolizidinone **14** (45 mg, 65%) as a yellow oil.: R_f_ (2:1 hexane/EtOAc) 0.55; 1H NMR (300 MHz, CDCl_3_) δ 7.50–7.13 (m, 10H), 6.67 (s, 1H), 6.58 (s, 1H), 4.90–4.82 (m, 2H), 4.00 (d, *J* =13.1 Hz, 2H), 3.92–3.73 (m, 8H), 3.58 (d, *J* = 13.1 Hz, 2H), 3.25 (dd, *J* = 11.4, 6.2 Hz, 1H), 3.20–3.04 (m, 1H), 2.88 (td, *J* = 11.4, 4.1 Hz, 1H), 2.64 (d, *J* = 14.3 Hz, 1H), 2.30–2.20 (m, 1H), 1.90−1.70 (m, 2H), 1.02 (d, *J* = 7.0 Hz, 3H); ^13^C NMR (75 MHz, CDCl_3_) δ 170.1, 148.1, 147.5, 140.5 (2C), 129.5, 129.2(4C), 128.8, 128.1(4C), 126.8(2C), 112.4, 107.6, 73.7, 58.5, 56.3, 55.8, 55.3, 53.7, 42.7, 39.2, 28.8, 25.6, 20.4, 13.2; ${\left[\alpha \right]}_{25}^{D}$ −74.7 (c 0.8, CHCl_3_) ); νmax (film) 2961, 2935, 1737, 1612, 1514, 1262, 1102, 732, 698 cm^−1^; ESI-HRMS calculated for C_31_H_37_N_2_O_4_ [M + H]^+^ 501.2753, found 501.2749.

### 3.3. (3R,11bS)-6,7-Dihydro-3-((S)-1-hydroxyethyl)-9,10-dimethoxy-3H-pyrido[2,1-a]isoquinolin-4(11bH)-one (**15**) 

To a solution of dibenzylamino-benzoquinolizidinone **14** (36 mg, 0.08 mmol) in CHCl_3_ (5 mL), *m*-CPBA (26 mg, 0.11 mmol) was added at 0 °C and the solution was stirred for 1 hour. The reaction was quenched with sat. aq. NaHCO_3_ (5 mL) and extracted with CH_2_Cl_2_ (3 × 5 mL). The combined organic layers were dried over anh. Na_2_SO_4_, filtered and evaporated under reduced pressure. The crude product was purified by column chromatography (silica gel, 20:1 CH_2_Cl_2_/MeOH) to give unsaturated lactam **15** (24 mg, quantitative) as a colorless oil.: R_f_ (20:1 CH_2_Cl_2_/MeOH) 0.45; ^1^H NMR (300 MHz, CDCl_3_) δ 6.68 (s, 1H), 6.82 (s, 1H), 6.50 (dd, *J* = 10.2, 2.1 Hz, 1H), 5.95 (d, *J* = 10.2 Hz, 1H), 4.96 (s, 1H), 4.70–4.60 (m, 1H), 4.05 (p, *J* = 7.0 Hz 1H), 3.85 (s, 3H), 3.84 (s, 3H), 3.22–3.01 (m, 2H), 2.94–2.2.85 (m, 1H), 2.65 (d, *J* = 14.3 Hz, 1H), 1.75 (brs, 1H), 1.40 (d, *J* = 7.0 Hz, 3H); 13C NMR (75 MHz, CDCl3) δ 161.6, 145.8, 144.7, 136.1, 126.3, 126.1, 123.3, 110.1, 105.2, 66.2, 54.4, 53.9, 42.2, 40.2, 27.3, 25.1, 18.8; ${\left[\alpha \right]}_{25}^{D}$ +32.8 (c 0.7, CHCl_3_); νmax (film) 3346, 2967, 2933, 1736, 1650, 1603, 1514, 1261, 1118, 749 cm^−1^; ESI-HRMS calculated for C_17_H_21_N_Na_O_4_ [M + Na]^+^ 326.1368, found 326.1359.

### 3.4. (1S,3S,11bS)-1-(Dibenzylamino)-2,3,6,7-tetrahydro-3-((S)-1-hydroxyethyl)-9,10-dimethoxy-1H-pyrido[2,1-a]isoquinolin-4(11bH)-one (**17**)

To a solution of glutarimide **12** (112 mg, 0.18 mmol) in dry toluene (10 mL), DIBALH (0.36 mL, 1M in toluene, 0.36 mmol) was added under an argon atmosphere at −78 °C. The solution was stirred for 1 hour and quenched with MeOH (2 mL). The resulting mixture was allowed to warm to room temperature and sat. aq. NaHCO_3_ (10 mL) was added. The mixture was extracted with EtOAc (3 × 10 mL) and the combined organic layers were dried over anh. Na_2_SO_4_, filtered and concentrated under reduced pressure to give hydroxylactam **16** as a yellow oil. This crude material was dissolved in dry CH_2_Cl_2_ (10 mL), and to this solution, TMSOTf (0.05mL, 0.30 mmol) was added at 0 °C under an argon atmosphere and stirred for 3 hours. The reaction was quenched with sat. aq. NaHCO_3_ (10 mL) and extracted with CH_2_Cl_2_ (3 × 10 mL). The combined organic layers were dried over anh. Na_2_SO_4_, filtered and evaporated under reduced pressure. The crude product was purified by flash chromatography (silica gel, 2:1 hexane/EtOAc) to give benzoquinolizidinone **17** (65 mg, 72%) as a yellow oil.: R_f_ (2:1 hexane/EtOAc) 0.43; ^1^H NMR (300 MHz, CDCl_3_) δ 7.48–7.17 (m, 10H), 6.89 (s, 1H), 6.68 (s, 1H), 4.25 (d, *J* = 7.0 Hz, 1H), 4.16–4.04 (m, 4H), 3.89–3.72 (m, 3H), 3.82(s, 3H), 3.76 (s, 3H), 3.36–3.15 (m, 2H), 3.00 (dt, *J* = 14.2, 7.0 Hz, 1H), 2.72 (dt, *J* = 14.2, 6.4 Hz, 1H), 2.42–2.30 (m, 1H), 2.15–2.05 (m, 1H), 1.86 (q, *J* = 7.0 Hz, 1H), 1.80 (brs, 1H), 1.28 (d, *J* = 7.0 Hz, 3H); ^13^C NMR (75 MHz, CDCl_3_) δ 173.6, 148.2, 147.0, 140.4(2C), 130.2, 129.2, 128.7(4C), 128.4(4C), 126.7(2C), 111.3, 109.0, 69.4, 57.9, 56.5, 56.1, 55.9, 55.4, 43.5, 42.6, 27.5, 26.4, 21.6, 19.9; ${\left[\alpha \right]}_{25}^{D}$ −14.3 (c 0.9, CHCl_3_); νmax (film) 3386, 2961, 2935, 1737, 1612, 1514, 1262, 1102, 732, 698 cm^−1^; ESI-HRMS calculated for C_31_H_37_N_2_O_4_ [M + H]^+^ 501.2753, found 501.2743.

### 3.5. (3S,11bR)-6,7-Dihydro-3-((R)-1-hydroxyethyl)-9,10-dimethoxy-3H-pyrido[2,1-a]isoquinolin-4(11bH)-one (ent-**15**)

To a solution of dibenzylamino-benzoquinolizidinone **17** (20 mg, 0.04 mmol) in CHCl_3_ (5 mL), *m*-CPBA (15 mg, 0.06 mmol) was added at 0 °C and the solution was stirred for 1 hour. The reaction was quenched with sat. aq. NaHCO_3_ (5 mL) and extracted with CH_2_Cl_2_ (3 × 5 mL). The combined organic layers were dried over anh. Na_2_SO_4_, filtered and evaporated under reduced pressure. The crude product was purified by flash chromatography (silica gel, 20:1 CH_2_Cl_2_/MeOH) to give tricyclic unsaturated lactam *ent*-**15** (12 mg, quantitative) as a colorless oil. All spectral data are identical to those of **15**.: R_f_ (20:1 CH_2_Cl_2_/MeOH) 0.45; ^1^H NMR (300 MHz, CDCl_3_) δ 6.68 (s, 1H), 6.82 (s, 1H), 6.50 (dd, *J* = 10.3, 2.2 Hz, 1H), 5.95 (d, *J* = 10.3 Hz, 1H), 4.96 (s, 1H), 4.70–4.60 (m, 1H), 4.05 (p, *J* = 7.0 Hz 1H), 3.85 (s, 3H), 3.84 (s, 3H), 3.22–3.01 (m, 2H), 2.94–2.2.85 (m, 1H), 2.65 (d, *J* = 14.3 Hz, 1H), 1.75 (brs, 1H), 1.40 (d, *J* = 7.0 Hz, 3H); ^13^C NMR (75 MHz, CDCl_3_) δ 161.6, 145.8, 144.7, 136.1, 126.3, 126.1, 123.3, 110.1, 105.2, 66.2, 54.4, 53.9, 42.2, 40.2, 27.3, 25.1, 18.8; ${\left[\alpha \right]}_{25}^{D}$ −44.8 (c 0.9, CHCl_3_); νmax (film) 3346, 2967, 2933, 1736, 1650, 1603, 1514, 1261, 1118, 749 cm^−1^; ESI-HRMS calculated for C_17_H_21_N_Na_O_4_ [M + Na]^+^ 326.1368, found 326.1356.

### 3.6. (3S, 5R)- and (3S,5S)-1-(3,4-Dimethoxyphenylethyl)-3-(dibenzylamino)- 5-(1-hydroxy-2-methylpropyl)piperidine-2,6-diones **18** and **19**

To a solution of glutarimide **7** (944 mg, 2.0 mmol) in dry THF (30 mL), LDA (1M solution, 3.0 mL, 3.0 mmol) was added at −78 °C under an argon atmosphere and the solution was stirred for 15 min. To this mixture, isobutyraldehyde (0.30 mL, 3.3 mmol) was added, and the resulting solution was stirred for an additional hour. The reaction was quenched with sat aq. NH_4_Cl (20 mL). The mixture was extracted with CH_2_Cl_2_ (3 × 20 mL). The combined organic layers were dried over anh. Na_2_SO_4_, filtered and evaporated under reduced pressure. Purification of the crude material by flash column chromatography (silica gel, 4:1 hexane/ethyl acetate) produced the aldol adducts (**18** = 479 mg, 44%, **19** = 250 mg, 23%) as yellow oils. 

**18**: R_f_ (2:1 hexane/ethyl acetate) 0.44; ^1^H NMR (300 MHz, CDCl_3_) δ 7.45–7.20 (m, 10H), 6.82–6.69 (m, 3H), 4.00 (t, *J* = 7.0 Hz, 1H), 3.91–3.81 (m, 4H), 3.88 (s, 3H), 3.80 (s, 3H), 3.60 (d, *J* = 13.1 Hz, 2H), 3.60–3.52 (m, 1H), 3.15 (dd, *J* = 7.0, 3.0 Hz, 1H), 2.70–2.81 (m, 3H), 2.18–2.01 (m, 1H), 1.71–1.80 (m, 1H), 1.52 (octet, *J* = 7.0 Hz, 1H), 0.85 (m, 6H); ^13^C NMR (75 MHz, CDCl_3_) δ 174.3, 173.0, 148.7, 147.6, 139.3(2C), 130.7, 128.6(4C), 128.3(4C), 127.2(2C), 121.1, 112.3, 110.9, 75.7, 55.8, 55.7, 55.1, 44.7, 40.6, 33.3, 33.6, 30.2, 26.2, 19.7, 14.8; ${\left[\alpha \right]}_{25}^{D}$ +31.0 (c 1.1, CHCl_3_); νmax (film) 3346, 2967, 2933, 1736, 1650, 1603, 1514, 1261, 1118, 749 cm^−1^; ESI-HRMS calculated for C_33_H_41_N_2_O_5_ [M + H]^+^ 545.3015, found 545.3019.

**19**: R_f_ (2:1 hexane/ethyl acetate) 0.32; ^1^H NMR (300 MHz, CDCl_3_) δ 7.42–7.18 (m, 10H), 6.80–6.70 (m, 3H), 4.05 (q, *J* = 7.0 Hz, 1H), 4.00–3.82 (m, 4H), 3.89 (s, 3H), 3.77 (s, 3H), 3.62 (d, *J* =13.1 Hz, 2H), 3.45 (dd, *J* = 6.2, 3.1 Hz, 1H), 2.79 (t, *J* = 7.1 Hz, 2H), 2.40 (ddd, *J* = 12.3, 7.1, 3.0 Hz, 1H), 1.81–1.71 (m, 2H), 1.75 (q, *J* = 8.2 Hz, 1H), 1.67 (brs, 1H), 1.05 (d, J = 7.0 Hz, 3H), 0.88 (d, *J* = 7.0 Hz, 3H); ^13^C NMR (75 MHz, CDCl_3_) δ 176.0, 172.7, 149.0, 147.7, 139.4(2C), 130.6, 128.5(4C), 128.4(4C), 127.2(2C), 121.1, 112.4, 111.0, 76.2, 59.0, 55.9, 55.7(2C), 54.9, 45.4, 41.0, 33.5, 29.8, 25.6, 19.9, 15.0; ${\left[\alpha \right]}_{25}^{D}$ −22.8 (c 0.9, CHCl_3_); νmax (film) 3340, 2965, 2933, 1740, 1650, 1603, 1514, 1261, 1118, 757 cm^−1^; ESI-HRMS calculated for C_33_H_41_N_2_O_5_ [M + H]^+^ 545.3015, found 545.3016.

### 3.7. (1S,3R,11bR)-1-(Dibenzylamino)-2,3,6,7-tetrahydro-3-((S)-1-hydroxy-2-methylpropyl)-9,10-dimethoxy-1H-pyrido[2,1-a]isoquinolin-4(11bH)-one (**21**)

To a solution of glutarimide **18** (530 mg, 0.97 mmol) in dry toluene (20 mL), DIBALH (1.95 mL, 1M in toluene, 1.95 mmol) was added under an argon atmosphere at –78 °C and the solution was stirred for 1 hour. The reaction was quenched with MeOH (3 mL). The resulting mixture was allowed to warm to room temperature and sat. aq. NaHCO_3_ (20 mL) was added. The mixture was extracted with EtOAc (3 × 15 mL) and the combined organic layers were dried over anh. Na_2_SO_4_, filtered and concentrated under reduced pressure to give hydroxylactam **20** as a light-yellow oil. This crude product was dissolved in dry CH_2_Cl_2_ (15 mL), and to this solution, TMSOTf (0.30 mL, 1.7 mmol) was added at 0 °C under an argon atmosphere. The reaction was stirred for 3 hours and quenched with sat. aq. NaHCO_3_ (15 mL) and extracted with CH_2_Cl_2_ (3 × 10 mL). The combined organic layers were dried over anh. Na_2_SO_4_, filtered and evaporated under reduced pressure. The crude product was purified by flash chromatography (silica gel, 2:1 hexane/ethyl acetate) to give benzoquinolizidinone **21** (384 mg, 75%) as a colorless oil.: R_f_ (2:1 hexane/ethyl acetate) 0.47; ^1^H NMR (300 MHz, CDCl_3_) δ 7.48–7.15 (m, 10H), 6.69 (s, 1H), 6.60 (s, 1H), 4.90–4.79 (m, 2H), 3.98 (d, *J* = 13.1 Hz, 2H), 3.90–3.80 (m, 2H), 3.85 (s, 6H), 3.79–3.61 (m, 1H), 3.57 (d, *J* = 13.1 Hz, 2H), 3.46 (dd, *J* = 8.2, 3.0 Hz, 1H), 3.24–3.01 (m, 2H), 2.85 (td, *J* = 12.4, 4.1 Hz, 1H), 2.35 (d, *J* = 12.4 Hz, 1H), 1.89–1.70 (m, 1H), 0.86 (d, *J* = 7.0 Hz, 3H), 0.70 (d, *J* = 7.0 Hz, 3H); ^13^C NMR (75 MHz, CDCl_3_) δ 171.1, 148.1, 147.5, 140.5(2C), 129.5, 129.2(4C), 128.8(4C), 128.10(2C), 112.4, 107.6, 73.7, 58.5, 56.4, 55.8, 55.3, 54.3, 53.7, 42.7, 39.2, 28.9, 28.6, 25.6, 20.4, 13.2 ; ${\left[\alpha \right]}_{25}^{D}$ −12.8 (c 1.0, CHCl_3_); νmax (film) 3340, 2965, 2933, 1740, 1650, 1603, 1514, 1261, 1118, 757 cm^−1^; ESI-HRMS calculated for C_33_H_41_N_2_O_4_ [M + H]^+^ 529.3066, found 529.3052.

### 3.8. (3R,11bS)-6,7-Dihydro-3-((S)-1-hydroxy-2-methylpropyl)-9,10-dimethoxy-3H-pyrido[2,1-a]isoquinolin-4(11bH)-one (**22**)

To a solution of tricyclic dibenzylamino-benzoquinolizidinone **21** (69 mg, 0.13 mmol) in CHCl_3_ (5 mL), *m*-CPBA (49 mg, 0.20 mmol) was added at 0 °C and the solution was stirred for 1 hour. The reaction was quenched with sat. aq. NaHCO_3_ (5 mL) and extracted with CH_2_Cl_2_ (3 × 5 mL). The combined organic layers were dried over anh. Na_2_SO_4_, filtered and evaporated under reduced pressure. The crude product was purified by flash chromatography (silica gel, 40:1 CH_2_Cl_2_/MeOH) to give tricyclic unsaturated lactam **22** (37 mg, 86%) as a colorless oil.: R_f_ (20:1 CH_2_Cl_2_/MeOH) 0.35; ^1^H NMR (300 MHz, CDCl_3_) δ 6.70 (s, 1H), 6.63 (s, 1H), 6.42 (dd, *J* = 10.3, 4.0 Hz, 1H), 5.90 (d, *J* = 10.3 Hz, 1H), 5.08 (s, 1H), 4.70–4.60 (m, 1H), 3.88 (s, 3H), 3.83 (s, 3H), 3.80–3.70 (m, 1H), 3.27–2.95 (m, 3H), 2.65 (d, *J* = 12.3 Hz, 1H), 1.95 (octet, *J* = 7.0 Hz, 1H), 1.73 (brs, 1H), 1.06 (d, *J* = 7.0 Hz, 6H); ^13^C NMR (75 MHz, CDCl_3_) δ 164.4, 148.2, 147.0, 138.3, 129.2, 128.6, 125.7, 112.7, 107.6, 76.0, 57.1, 56.4, 55.9, 43.2, 40.2, 29.7, 27.1, 20.4, 14.5; ${\left[\alpha \right]}_{25}^{D}$ +83.7 (c 1.9, CHCl_3_); νmax (film) 3247, 2971, 2933, 1602, 1275, 769 cm^-1^; ESI-HRMS calculated for C_19_H_26_NO_4_ [M + H]^+^ 332.1862, found 332.1861.

### 3.9. (1S,3S,11bS)-1-(Dibenzylamino)-2,3,6,7-tetrahydro-3-((R)-1-hydroxy-2-methylpropyl)-9,10-dimethoxy-1H-pyrido[2,1-a]isoquinolin-4(11bH)-one (**24**)

To a solution of glutarimide **19** (300 mg, 0.55 mmol) in dry toluene (20 mL), DIBALH (1.1 mL, 1M in toluene, 1.1 mmol) was added under an argon atmosphere at −78 °C and the solution was stirred for 1 hour. The reaction was quenched with MeOH (3 mL). The resulting mixture was allowed to warm to room temperature and sat. aq. NaHCO_3_ (20 mL) was added. The mixture was extracted with EtOAc (3 × 15 mL) and the combined organic layers were dried over anh. Na_2_SO_4_, filtered and concentrated under reduced pressure to give hydroxylactam **23** as a light-yellow oil. This crude product was dissolved in dry CH_2_Cl_2_ (15 mL), and to this solution, TMSOTf (0.40 mL, 2.2 mmol) was added at 0 °C under an argon atmosphere. The reaction was stirred for 3 hours and quenched with sat. aq. NaHCO_3_ (15 mL) and extracted with CH_2_Cl_2_ (3 × 10 mL). The combined organic layers were dried over anh. Na_2_SO_4_, filtered and evaporated under reduced pressure. The crude product was purified by flash chromatography (silica gel, 2:1 hexane/ethyl acetate) to give benzoquinolizidinone **24** (238 mg, 82%) as a colorless oil.: R_f_ (2:1 hexane/ethyl acetate) 0.35; ^1^H NMR (300 MHz, CDCl_3_) δ 7.48–7.12 (m, 10H), 7.06 (s, 1H), 6.58 (s, 1H), 4.62–4.45 (m, 2H), 4.10 (d, *J* = 13.1 Hz, 2H), 3.83 (s, 3H), 3.78 (s, 3H), 3.78 (d, *J* = 13.1 Hz, 2H), 3.60–3.45 (m, 2H), 3.30–3.00 (m, 3H), 3.53 (dd, *J* = 7.0, 3.0 Hz, 1H), 3.20–3.00 (m, 3H), 2.73–2.70 (m, 1H), 2.52–2.39 (m, 1H), 2.10–1.91 (m, 2H), 1.90–1.70 (m, 1H), 1.07 (d, *J* = 7.0 Hz, 3H), 1.00 (d, *J* = 7.0 Hz, 3H); ^13^C NMR (75 MHz, CDCl_3_) δ 172.4, 147.9, 146.9, 140.5(2C), 131.8, 128.5(4C), 128.2, 127.8(4C), 126.7(2C), 111.4, 110.2, 80.1, 57.8, 57.7, 55.9(2C), 55.3(2C), 42.8, 40.4, 30.0, 29.2, 27.3, 20.5, 14.6; ${\left[\alpha \right]}_{25}^{D}$ −8.0 (c 0.9, CHCl_3_); νmax (film) 3373, 2959, 2928, 1671, 1262, 732, 698 cm^−1^; ESI-HRMS calculated for C_33_H_41_N_2_O_4_ [M + H]^+^ 529.3066, found 529.3048.

### 3.10. (3S,11bR)-6,7-Dihydro-3-((R)-1-hydroxy-2-methylpropyl)-9,10-dimethoxy-3H-pyrido[2,1-a]isoquinolin-4(11bH)-one (ent-**22**)

To a solution of tricyclic dibenzylamino-benzoquinolizidinone **24** (222 mg, 0.42 mmol) in CHCl_3_ (10 mL), *m*-CPBA (148 mg, 0.60 mmol) was added at 0 °C and the solution was stirred for 1 hour. The reaction was quenched with sat. aq. NaHCO_3_ (10 mL) and extracted with CH_2_Cl_2_ (3 × 10 mL). The combined organic layers were dried over anh. Na_2_SO_4_, filtered and evaporated under reduced pressure. The crude product was purified by flash chromatography (silica gel, 40:1 CH_2_Cl_2_/MeOH) to give unsaturated lactam *ent*-**22** (134 mg, 96%) as a colorless oil. All spectral data are identical to those of **22**.: R_f_ (20:1 CH_2_Cl_2_/MeOH) 0.35; ^1^H NMR (300 MHz, CDCl_3_): δ 6.70 (s, 1H), 6.63 (s, 1H), 6.42 (dd, *J* = 10.3, 4.0 Hz, 1H), 5.90 (d, *J* = 10.3 Hz, 1H), 5.08 (s, 1H), 4.70–4.60 (m, 1H), 3.88 (s, 3H), 3.83 (s, 3H), 3.80–3.70 (m, 1H), 3.27–2.95 (m, 3H), 2.65 (d, *J* = 12.3 Hz, 1H), 1.95 (octet, *J* = 7.0 Hz, 1H), 1.73 (brs, 1H), 1.06 (d, *J* = 7.0 Hz, 6H); ^13^C NMR (75 MHz, CDCl_3_) δ 164.4, 148.2, 147.0, 138.3, 129.2, 128.6, 125.7, 112.7, 107.6, 76.0, 57.1, 56.4, 55.9, 43.2, 40.2, 29.7, 27.1, 20.4, 14.5; ${\left[\alpha \right]}_{25}^{D}$ −79.5 (c 1.8, CHCl_3_); νmax (film) 3247, 2971, 2933, 1602, 1275, 769 cm^−1^; ESI-HRMS calculated for C_19_H_26_NO_4_ [M + H]^+^ 332.1862, found 332.1861.

### 3.11. (3S, 5S)-1-(3,4-Dimethoxyphenylethyl)-3-(dibenzylamino)- 5-((R)-1-hydroxyethyl) piperidine-2,6-dione (**26**)

To a solution of glutarimide **7** (473 mg, 1.0 mmol) in dry THF (10 mL), NaHMDS (2M solution, 1.50 mL, 3.0 mmol) was added at −78 °C under an argon atmosphere and the mixture was stirred for 15 min. To this mixture, acetaldehyde (0.17 mL, 3.0 mmol) was added, and the mixture was allowed to warm to room temperature while being stirred for 3 hours. The reaction was quenched by sat aq NH_4_Cl (10 mL) into the reaction. The mixture was extracted with CH_2_Cl_2_ (3 × 30 mL). The combined organic layers were dried over anh. Na_2_SO_4_, filtered and evaporated under reduced pressure to give the crude aldol adducts. This mixture was dissolved in dry CH_2_Cl_2_ (10 mL), and to the solution, 2,6-lutidine (0.14 mL, 1.2 mmol) and TBSOTf (0.34 mL, 1.5 mmol) were added at 0 °C under an argon atmosphere. The solution was stirred for 3 hours and quenched with sat. aq. NaHCO_3_ (10 mL) and extracted with CH_2_Cl_2_ (3 × 10 mL). The combined organic layers were dried over anh. Na_2_SO_4_, filtered and evaporated under reduced pressure. The crude product was purified by column chromatography (silica gel, 4:1 hexane/EtOAc) to give silyl ethers (**26** = 302 mg, 48%, **12** = 151 mg, 24%) as yellow oils.

**26**: R_f_ (4:1 hexane/EtOAc) 0.58; ^1^H NMR (300 MHz, CDCl_3_) δ 7.40–7.10 (m, 10H), 6.69–6.58 (m, 3H), 4.51–4.40 (m, 1H), 4.00–3.80 (m, 2H), 3.82 (d, *J* = 13.1 Hz, 2H), 3.79 (s, 3H), 3.66 (s, 3H), 3.55 (d, *J* = 13.1 Hz, 2H), 3.35 (dd, *J* = 12.3, 7.0 Hz, 1H), 2.90–2.82 (m, 1H) 2.68 (t, *J* = 7.0 Hz, 2H), 2.41–2.30 (m, 1H), 2.20–2.10 (m, 1H), 0.97 (d, *J* = 7.0 Hz, 3H), 0.82 (s, 9H), −0.01 (s, 3H), −0.07 (s, 3H); ^13^C NMR (75 MHz, CDCl_3_) δ 173.2, 172.3, 148.8, 147.7, 139.5, 130.7(2C), 128.4(4C), 128.4(4C), 127.2(2C), 112.3, 110.9, 66.7, 59.8, 55.8(2C), 55.7(2C), 54.8, 49.0, 40.8, 33.6, 29.7, 25.7(3C), 21.0, 18.7, 18.0, −4.6, −5.2; ${\left[\alpha \right]}_{25}^{D}$ −31.8 (c 1.0, CHCl_3_); νmax (film) 2950, 2929, 1724, 1669, 1515, 1260, 1028, 775 cm^−1^; ESI-HRMS calculated for C_37_H_50_N_2_NaO_5_Si [M + Na]^+^ 653.3387, found 653.3390.

### 3.12. (1S,3S,11bS)-1-(Dibenzylamino)-2,3,6,7-tetrahydro-3-((R)-1-hydroxyethyl)-9,10-dimethoxy-1H-pyrido[2,1-a]isoquinolin-4(11bH)-one (**28**)

To a solution of glutarimide **26** (530 mg, 0.97 mmol) in dry toluene (10 mL), DIBALH (1.95 mL, 1M in toluene, 1.95 mmol) was added under an argon atmosphere at −78 °C and the solution was stirred for 1 hour. The reaction was quenched with MeOH (2 mL). The resulting mixture was allowed to warm to room temperature and sat. aq. NaHCO_3_ (15 mL) was added. The mixture was extracted with EtOAc (3 × 5 mL) and the combined organic layers were dried over anh. Na_2_SO_4_, filtered and concentrated under reduced pressure to give hydroxylactam **27** as a light-yellow oil. This crude product was dissolved in dry CH_2_Cl_2_ (10 mL), and to the solution, TMSOTf (0.21 mL, 1.2 mmol) was added at 0 °C under an argon atmosphere. The solution was stirred for 3 hours and quenched with sat. aq. NaHCO_3_ (15 mL) and extracted with CH_2_Cl_2_ (3 × 10 mL). The combined organic layers were dried over anh. Na_2_SO_4_, filtered and evaporated under reduced pressure. The crude product was purified by flash chromatography (silica gel, 2:1 hexane/ethyl acetate) to give benzoquinolizidinone **28** (393 mg, 81%) as a colorless oil.: R_f_ (2:1 hexane/EtOAc) 0.72; ^1^H NMR (300 MHz, CDCl_3_) δ 7.50–7.14 (m, 10H), 6.72 (s, 1H), 6.68 (s, 1H), 4.55 (d, *J* = 8.0 Hz, 1H), 4.20–4.05 (m, 3H), 4.08 (d, *J* = 13.1 Hz, 2H), 3.85 (s, 3H), 3.82 (s, 3H), 3.81 (d, *J* = 13.1 Hz, 2H), 3.39–3.30 (m, 1H), 3.29–3.12 (m, 1H), 2.90–2.64 (m, 2H), 2.40–1.80 (m, 3H), 1.35 (d, *J* =7.0 Hz, 3H); ^13^C NMR (75 MHz, CDCl_3_) δ 171.2, 148.2, 147.0, 140.5(2C), 130.1, 129.3(4C), 128.7, 128.6(4C), 126.8(2C), 111.3, 108.9, 66.9, 56.5, 56.2, 56.1, 55.9, 55.3, 43.6, 41.9, 28.1, 26.4, 21.6, 14.2; ${\left[\alpha \right]}_{25}^{D}$ −64.4 (c 0.9, CHCl_3_); νmax (film) 3386, 2961, 2934, 1629, 1514, 1257, 1102, 732, 698 cm^−1^; ESI-HRMS calculated for C_31_H_37_N_2_O_4_ [M + H]^+^ 501.2753, found 501.2753.

## 4. Conclusions

In conclusion, we have devised a synthetic route from L-glutamic acid that led to both enantiomers of functionalized benzoquinolizidinone systems, making this synthesis enantiodivergent. The two enantiomers were derived from different modes of cyclic stereo-control. In this manner, we can obtain the benzoquinolizidinone system with *R* or S configuration at the C11b stereogenic center from the kinetic (*anti*-) or thermodynamic (*syn*-) aldol adduct, respectively. It is conceivable to make this approach enantioselective by optimization of the reaction conditions for either kinetic or thermodynamic products. The attempts to master the control of the diastereoselective glutarimide aldol reaction as well as conversion of the synthetic benzoquinolizidinone cores to natural alkaloids and their derivatives are ongoing.

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
