# Peer review of "Enantiodivergent Synthesis of Benzoquinolizidinones from L-Glutamic Acid"

_molecules, 2021, doi:10.3390/molecules26195866_

Round 1

Reviewer 1 Report

This is an interesting paper describing a synthetic approach to the title alkaloids from glutamic acid. The work has been carried out to a good standard and the paper is clearly written with good quality structures. It can be acceoted for publication subject to the following, mostly minor, matters being attended to:

Line 26 make ref 4 not bold.

Line 32 - Napieralski should end in i not y

Line 77 - make cpd no. 10 bold

Lines 92 and 115 - in 1H and 13C the numbers should be superscript

Line 94 - It is stated that the optical rotations are "similar in magnitude". Given that they should in principle be the same, the discrepancy between 32.8 and 44.8 is rather large. If possible the authors should comment on why this might be. For example are there optically active impurities present in one or other material? (For 22 there is much closer agreement)

Line 122, 132, 136, 140, 146 etc. - in N-acyliminium the N should be italic. Please check this all through.

Also in lines 128, 136, 140, 141, 147 - R and S should be italic. Again check this throughout the paper.

Line 151 - the term "materials and methods" is one used by biologists. For chemistry "experimental section" is preferable

Line 164 - the cell length being given as 0.9998 dm is ridiculous. How have the authors measured that it is 20 microns short of 1.0 dm and what is the accuracy of such measurement? This needs to be simply rounded off to 1 dm!

Lines 184, 204, 216, 227, 232 etc. - the compound numbers should be bold for emphasis wherever they appear, not just where the data is listed. Also in the headings this would be helpful so lines 169, 203, 226, 242 etc. Please check this throughout.

Line 294, 342 and 380 we have "o" for the H NMR signal which I assume means octet. Since this is not a common abbreviation it should rather be written out as "octet" in these three places.

Funding - delete the "Please add"

References - in both ref 4 and 8 the first word of the journal abbreviation should be in italics; Ann. Neurosci. and Adv. Heterocycl. Chem. respectively

Supplementary material - largely satisfactory but in the 13C NMR for 17, 18, 19 and 28 a lot of minor impurity signals are labelled. If possible please re-label with the main compound peaks only singled out as has been done for the other 13C spectra. Also remove the extra title in the middle of the cpd 17 page.

Author Response

Thank you very much for your thorough and thoughtful review of the manuscript. I am happy that you find our work interesting. I have made corrections according to your suggestion for the writing part. Also, 13C NMR spectra for Compounds 18 and 19 were replaced with cleaner ones. The integrations of 1H NMR and labeling of 13C NMR peaks are checked so that they correspond to the structure of the compounds.

Regarding the difference in optical rotations of 15 and ent-15, some impurity can be observed in the aliphatic region of the 1H NMR of 15. Even though I doubt that this impurity is optically active but it can affect the actual mass of the compound 15 that was active in the measurement.  When the mass of 15 combined with the impurity was used for the calculation of the [a]D this can lead to the value that is lower than expected. I suspect that this impurity came from the solvents that we used for the purification and we have not been successful to remove it for this particular compound. I made a note about this issue in reference # [15] in the manuscript.

Best regards,

Punlop Kuntiyong

Reviewer 2 Report

Kuntiyong’s group realized enantiodivergent synthesis of benzoquinolizidinones from L- 2 glutamic acid. This topic is interesting. The article is suitable for the journal. After the following comments were addressed, the manuscript can be published.

  1. There should be a caption after Scheme 1, Scheme 2 and so on.
  2. Scheme 1 is not marked in the text.
  3. it is better to added another scheme to display the reported procedures for synthesis of benzoquinolizidine frameworks more cleaer.

Author Response

Thank you very much for your suggestion. I am happy that you find our work interesting. According to your suggestion, I have included a scheme showing examples of previously reported syntheses of benzoquinolizidinone. Even though the reported methods are efficient and deliver the desired benzoquinolizidinone in a few steps, they are specific to the enantiomer of the chiral starting material tyrosine and the L-proline-derived chiral catalyst. Our method provides an alternative that is enantiodivergent.  

Also, all figure and schemes are now captioned.

Best regards,

Punlop Kuntiyong

Author Response

     Thank you so much for your review. I understand your critique in regard to the lack of novelty of the synthesis. Our intention is not developing new reactions or reagents for organic synthesis but rather manipulation of classical methods to achieve asymmetric synthesis. This manuscript is intended for the special issue “Synthesis of Tetrahydroisoquinoline and Protoberberine Derivatives”. In the web page for this Molecules’ special issue, the Editor listed ‘The stereochemical modification of the traditional, classical methods of the synthesis of the tetrahydroisoquinoline system’ as one of the strategies.  I believe our approach falls into this category. That is the main reason why we submitted this manuscript for the consideration of publication in this special issue. Our research group’s main strategy for asymmetric synthesis of piperidine and pyrrolidine containing alkaloids is utilization of chiral glutarimide and succinimide derived from L-glutamic and L-aspartic acid, respectively. We use manipulation of diastereoselective and regioselective reactions of these cyclic imides to control the stereochemical outcomes of the synthesis. Enantiodivergent synthesis has become our focus recently.

Thank you for your suggestion of conditions that can affect the diastereoselectivity of the aldol reactions. We will try out different conditions in our future work. It is our intention to improve the process so that the synthesis can be both enantiodivergent and enantioselective.

As for the NMR data, I have corrected the writing so that the coupling constants are reported with 1 decimal.  1H NMR spectrum of Cpd 28 and 13C NMR spectra for Cpds 18 and 19 are replaced with cleaner spectra. The integrations in the 1H NMR spectra for the TBS groups are now accounted for. 

  Best regards,

 Punlop Kuntiyong

Round 2

Reviewer 3 Report

The authors have well addressed concerns raised by the revieweres. This paper is now suitable for the publication in Molecules.